# Cure the Incurable? Recent Breakthroughs in Immune Checkpoint Blockade for Hepatocellular Carcinoma

**DOI:** 10.3390/cancers13215295

**Published:** 2021-10-22

**Authors:** Pei-Yi Chu, Shih-Hsuan Chan

**Affiliations:** 1National Institute of Cancer Research, National Health Research Institutes, Tainan 704, Taiwan; chu.peiyi@msa.hinet.net; 2College of Medicine, National Chung Hsing University, Taichung 402, Taiwan; 3Department of Pathology, Show Chwan Memorial Hospital, Changhua 500, Taiwan; 4School of Medicine, College of Medicine, Fu Jen Catholic University, Taipei 242, Taiwan; 5Department of Health Food, Chung Chou University of Science and Technology, Changhua 510, Taiwan; 6Graduate Institute of Integrated Medicine, China Medial University, Taichung 402, Taiwan

**Keywords:** hepatocellular carcinoma, checkpoint inhibitors, CTLA-4, PD1, PDL1

## Abstract

**Simple Summary:**

Liver cancer is one of the most devastating human malignancies worldwide, especially in Asia, where over 70% of new cases are diagnosed. Most liver cancers are classified as hepatocellular carcinoma (HCC). HCC patients usually present at an advanced stage and have very poor prognosis due to the inaccessibility of curative treatments and ineffective systemic therapies. Fortunately, recent clinical trials using checkpoint inhibitor (ICI) immunotherapy have obtained promising results to significantly prolong the overall survival of patients and improve quality of life. In this review, we summarize the recent efforts of ICI-related clinical trials and also point out the future directions of ICI-related immunotherapy for HCC.

**Abstract:**

HCC usually arises from a chronic inflammation background, driven by several factors including fatty liver, HBV/HCV viral infection and metabolic syndrome. Systemic treatment for advanced HCC remains disappointing due to its strong resistance to chemotherapy and even to tyrosine kinase inhibitors (TKIs). Recently, the use of ICI therapy has revolutionized the systemic treatment of advanced HCC. For the first time, clinical trials testing ICIs, anti-CTLA-4 and anti-PD1/PDL1 reported a survival benefit in patients with sorafenib resistance. However, it took four more years to find the right combination regimen to use ICI in combination with the anti-angiogenic agent bevacizumab to substantially prolong overall survival (OS) of patients with advanced HCC after sorafenib. This review provides a comprehensive history of ICI therapy in HCC, up-to-date information on the latest ICI clinical trials, and discusses the recent development of novel ICIs that would potentially lead to a new checkpoint blockade therapy for advanced HCC.

## 1. Introduction

Hepatocellular carcinoma is the most prevalent form of liver cancer in the world [1]. Annually, HCC affects approximately 900,000 individuals, and over 70% of new cases are diagnosed in Asia [2]. The etiology of HCC is complicated due to the multiple risk factors involved [3]. HCC usually arises from a background of chronic liver disease caused by alcohol abuse, metabolic syndrome, hepatitis B or C infections and/or aflatoxin exposure that eventually scars the liver parenchyma, leading to the irreversible condition of liver cirrhosis and the subsequent development of HCC [4]. Most HCC patients are diagnosed with advanced disease, and the majority of them are unresectable due to the early dissemination of cancer cells inside the liver. HCC tends to grow near blood vessels such as the portal vein or hepatic vein, which makes surgery an impossible task. α-Fetoprotein (AFP) is commonly used as a serum biomarker for early detection of HCC and also for evaluation of the prognosis and monitoring of response to therapy [5,6]. Systemic chemotherapy for many years is ineffective and therefore not an option for HCC, due to the highly resistant nature of HCC. Locoregional therapies including percutaneous ablation, transarterial chemoembolization (TACE) and radioembolization serve as the main alternative therapies, but largely depend on tumor location, burden and other complications [7]. Ablation is the first line option over surgery for unresectable HCC; however, treatment outcome is still disappointing and recurrence is often seen. In 2008, a tyrosine kinase inhibitor (TKIs), sorafenib, was the first approved targeted therapy to be used as a first-line drug to treat advanced HCC [8]. Subsequently, three additional TKIs, lenvatinib, regorafenib and cabozantinib, have been approved and become available for use in first-line and second-line settings and found to provide beneficial effects and prolonged survival [9,10]. Despite of the application of TKI therapy, median survival with advanced HCC remains unsatisfactory at less than two years [7,11]. Recently, ramucirumab, a therapeutic monoclonal antibody drug that acts against vascular endothelial growth factor (VEGF) receptor 2, has shown significant survival benefits in patients with increased AFP (>400 ng/mL) after sorafenib [12]. A VEGF-neutralizing antibody, bevacinumab, has also proven effective in a single-agent phase II trial in HCC patients. However, serious bleeding complications were observed in 11% [13].

Immune surveillance plays an important role in identifying and eliminating normal cells that become malignant. As when fighting invading pathogens, the innate and adaptive immune systems work together to form an anti-tumor army to destroy cancer cells. However, the interplay between cancer cells, stromal cells, and infiltrating immune cells eventually creates an immunosuppressive tumor microenvironment (TME) that leads to immune evasion, which was previously considered impossible to reverse. There are two main aspects explaining the formation of an immunosuppressive TME. First, as HCC progresses, cancer cells recruit immune cells such as myeloid-derived suppressor cells (MDSCs) or M2 tumor-associated macrophages (TAMs) and FoxP3^+^ regulatory T cells (T_reg_), which are known to help the tumor grow better, by secreting chemokines, cytokines and growth factors to form a tumor-promoting niche. As a result, the accumulation of tumor-promoting immunes cells eventually comprises an area of anti-tumor immunity. Second, upregulation of co-inhibitory molecules such as immune checkpoint ligand PDL1 and increased expression of tolerance-related enzymes such as indoleamine 2,3-deoxygenase (IDO) and arginase-1 in cancer cells or tumor-infiltrating immune cells also contribute to the formation of immunosuppressive TME. In addition, downregulation of tumor-associated antigens (TAAs), also known as tumor antigen escape [14,15], and reduced recognition of TAAs by immune cells through alterations in the antigen-processing machinery both play a significant role in the promotion of tumor progression [16]. Therefore, immunotherapies aiming to reverse and overcome the immunosuppressive TME in order to effectively enhance the activity of tumor-killing immune cells point out the future direction of HCC therapy. 

## 2. Tumor Microenvironment of HCC

The liver is the organ responsible for the detoxification of gut-derived blood and systemic circulation. Therefore, the frequent exposure of liver cells to food antigens and to microbial products generated by gut bacteria shapes the dynamic complexities of a liver microenvironment that fosters immune tolerance [17]. This tolerogenic milieu is maintained by liver antigen-presenting cells (APCs), including resident kuffer cells (KCs), sinusoidal endothelial cells (SECs) and stellate cells (SCs), through the secretion of an array of immunosuppressive cytokines, chemokines and growth factors [1]. Kuffer cells are known to express immunosuppressive cytokine IL-10 and IDO, and prostaglandins to promote T_reg_ activation [18,19]. Myeloid-derived suppressor cells (MDSCs), dendritic cells (DCs) and regulatory T cells also produce IL-10 to attenuate the ability of APCs to stimulate T cells and to promote PDL1 expression in monocytes [20]. TGF-β is a well-known soluble factor that attenuates the anti-tumor response by inhibiting the activation of dendritic cells (DCs) [21] and polarizing macrophages towards the M2 phenotype [22], as well as inducing T_reg_ cell activation [23]. In HCC TME, TGF-β is mainly produced by cancer cells, T_reg_ cells and macrophages [17]. High serum TGF-β has been linked to poor prognosis of patients with HCC after sorafenib treatment [24]. SCs-derived hepatocyte growth factor (HGF) also promotes the infiltration and accumulation of MDSCs and T_reg_ cells inside the TME [25,26]. Vascular endothelial growth factor (VEGF) produced by cancer cells or MDSCs plays an essential role in promoting angiogenesis, leading to the formation of an abnormal tumor vasculature that not only serves as a barrier for cytotoxic T lymphocytes (CTLs) but disables them by expressing PDL1 and Fas ligand [27,28,29]. Aside from its role in inducing abnormal tumor vasculature, Courau, et al. showed that VEGF and TGF-β could cooperatively foster immunotolerant TME by blunting the antigen-presenting functions of DC and generating MDSCs [30]. 

## 3. Current and Ongoing Strategies of Immune Checkpoint Blockade for HCC

Co-inhibitory molecules expressed by the effector lymphocytes fall into a category of immune checkpoint which serves the purpose of preventing overactivation of lymphocytes upon the engagement of APCs like DCs or macrophages [31]. To evade immune surveillance, tumor cells exploit this mechanism to express the corresponding ligands of co-inhibitory molecules to blunt effector T cells or macrophages. Program death 1 (PD1), cytotoxic T lymphocyte-associated antigen 4 (CTLA4), T cell immunoglobulin and mucin domain containing-3 (TIM3), lymphocyte-activation gene 3 (LAG3), and siglec-10 are among the most extensively studied immune checkpoints [32,33]. PD1 has been found to be expressed by activated T cells, nature killer (NK) cells, MDSCs, monocytes and DCs [34,35,36]. Its corresponding ligand, PDL1, has been found in tumor cells, stromal cells, and myeloid lineage cells like macrophages and DCs [36]. CTLA4 is predominantly expressed by T_reg_ cells and only upregulated in activated T cells [37,38]. In addition, the sialic acid-binding immunoglobulin (Ig)-like lectins 10 (siglec-10) has recently been identified as a new class of co-inhibitory molecule expressed by macrophages to interact with its corresponding ligand, CD24, expressed on the tumor surface [33,39]. 

Approaches taken to blocking the above-mentioned immune checkpoint receptor/ligand interactions have been considered as a form of immune checkpoint blockade therapy, illustrated in Figure 1. 

### 3.1. Monotherapy 

Here, we summarize the current milestones regarding recent clinical trials testing ICIs as a potential systemic treatment for advanced HCC (Figure 2). The first immune checkpoint inhibitor (ICI), ipilimumab, the anti-CTLA-4 monoclonal antibody (mAb), was approved by the U.S. Food and Drug Administration (FDA) in March 2011 for the treatment of patients with advanced melanoma [40]. In 2013, a pilot clinical trial involving 20 patients with advanced HCC and a background of chronic hepatitis C virus (HCV) infection who received tremelimumab treatment, another anti-CTLA-4 mAb, showed promising results in terms of safety, antitumor and antiviral activity [41]. This encouraging result has led to the approval of another ICI, the PD1 inhibitor nivolumab, for the treatment of patients after sorafenib failure in the CheckMate 040 phase I/II clinical trial. In the CheckMate 040 trial (*n* = 262), durable objective response rate (ORR), defined as the sum of complete (CR) and partial (PR) response rates, was observed in 20% (95% CI 15–26) in patients treated with nivolumab 3 mg/kg in the dose-expansion phase and 15% (95% CI 6–28) in the dose-escalation phase, The median overall survival (OS) in sorafenib-experienced patients was 16.5 months. Further, the two-year survival rate among the responders was over 80% [42]. Soon after, nivolumab was approved by the FDA as a second-line systemic therapy for advanced HCC after sorafenib. Another PD1 inhibitor, pembrolizumab. was also shown effective and tolerable in patients with advanced HCC after sorafenib in a non-randomized, open-label phase II trial. ORR was recorded in 17% 95% CI 11–26) of patients previously treated with sorafenib [43]. Given the consistent results in terms of anti-tumor activity and safety from different PD1 mAb, a randomized, double-blind, phase III trial (KEYNOTE 240) (*n* = 413) testing pembrolizumab versus placebo after sorafenib failure was launched in a second-line setting. Median OS was 13.9 months (95% CI 11.6–16) in the pembrolizumab treatment group and 10.36 months (95% CI 8.3–13.5) in the placebo group, and a statistically significant survival benefit was observed (Hazard ration HR 0.78; *p* = 0.0238) in the final analysis [44]. Although, OS and progression-free survival (PFS) did not reach the prespecified criteria, the results were in line with KEYNOTE 224, indicating a favorable risk-to-benefit ratio in the pembrolizumab group. Table 1 summarizes current important phase III trials involving ICI therapy as a major treatment modality.

In 2019, the phase III study CheckMate 459 compared the clinical efficacy and safety of nivolumab with sorafenib in 743 patients with treatment-naïve advanced HCC in a first-line setting [52]. Patients were randomized at a 1:1 ratio to either the nivolumab or sorafenib arm, with a follow-up of 22.8 months. ORR was 15% for patients who received nivolumab and 7% for those who received sorafenib. Prolonged survival was observed in patients after nivolumab, with a median OS of 16.4 months (95% CI 13.9–18.4), compared to patients after sorafenib with median OS of 14.7 months (95% CI 11.9–17.2) (HR 0.85; 95% CI 0.72–1.02; *p* = 0.0752). Although the predefined threshold of statistical significance for OS was not met (HR 0.84, *p* = 0.0419) [53], other end point parameters favored nivolumab over sorafenib, with a better disease control (median 7.5 months versus 5.7 months) and safety profile with fewer treatment-related adverse effects (22% versus 49%). Long-term follow-up of CheckMate 459 has also shown that first-line nivolumab monotherapy demonstrated clinical meaningful benefit in patients with advanced HCC at a minimum follow-up of 33.6 months, and that nivolumab had a more favorable safety profile and better preservation of liver function over the course of treatment as compared with sorafenib [53]. Given the promising results from the initial attempts with ICIs in HCC, different mAb targeting PD1/PDL1 have been developed and approved by the U.S. FDA for the treatment of patients with advanced HCC, and a number of combination strategies have been considered and tested in current clinical trials [47,48,49,50,51]. (Table 1). 

### 3.2. Combination Therapy 

In the CheckMate 040 randomized, open-label, multicohort phase I/II trial testing nivolumab plus ipilimumab in patients with unresectable HCC after sorafenib, patients were randomized at a 1:1:1 ratio to three treatment arms: (A) nivolumab 1 mg/kg plusipilimumab 3 mg/kg every three weeks (4 doses in total) followed by nivolumab 240 mg; (B) nivolumab 3 mg/kg plus ipilimumab 1 mg/kg followed by nivolumab 240mg; or (C) nivolumab 3 mg/kg every two weeks plus ipilimumab 1 mg/kg every 6 weeks. Although this study reported higher rates of adverse events with combination therapy of nivolumab and ipilimumab than previously reported in nivolumab monotherapy [42], nivolumab plus ipilimumab had manageable safety, promising objective response rate, and durable response with 32% (16 of 50 patients) ORRs in arm A, 31% (15 of 49 patients) in arm B, and 31% (15 of 49) in arm C. Patients in arm A had the best median OS of 22.8 months, and highest 30-month survival rate of 44% with better health improvement compared to that in the other two arms [45]. Four patients in arm A (4 of 49 patients) had a complete response, as did three patients in arm B (3 of 49 patients) [45]. Based on this clinical finding, the arm A regimen received accelerated approval by the U.S. FDA and further investigation of this regimen as first-line treatment for HCC is underway. Moreover, these results showed that the PD1/PDL1 and CTLA-4 pathways have a distinct role in modulating immune activity [54], and also confirmed that CTLA-4 blockade is effective in HCC, as in previous clinical trials [41,55]. Unlike PD1 and PDL1 inhibitors, dosing and timing seem crucial for anti-CTLA-4 therapy.

Another clinical trial attempting to target both PD1 and CTLA-4 with the different mAbs durvalumab (a PD1 inhibitor) and tremelimumab (a CTLA-4 inhibitor) in unresectable HCC also showed promising results; the combination regimen (tremelimumab 300 mg plus durvalumab 1,500 mg followed by durvalumab 1500 mg once every four weeks) had the best benefit-to-risk profile, with one patient having a complete response (1.3%, 1/75 patients), 17 patients a partial response (22.7%, 17/75 patients), and 16 patients stable (21.3%, 16/75 patients) [56]. Based on the phase I/II results, the randomized phase III HIMALAYA is now under way to assess the efficacy and safety of durvalumab plus tremelimumab both in combination and as monotherapy versus sorafenib in treatment-naïve patients with unresectable HCC [47]. 

Although experiences from ICI clinical trials show that no more than 20% of patients respond to ICI monotherapy, the treatment efficacy and ORRs of ICI therapy are still far better than previous TKI-related targeted therapies [57]. Moreover, combined therapy using nivolumab and ipilimumab has displayed the best ORR (32%) compared to any form of systemic treatment for HCC.

### 3.3. Combination Therapy of ICIs and VEGF Inhibitor

VEGF secreted by cancer cells is important in fostering an immunosuppressive TME, not only because of its role in recruiting endothelial cells to promote tumor angiogenesis (leading to the formation of abnormal tumor vasculature), but also for its suppressive role in inhibiting the differentiation and maturation of DCs. [29]. In addition, VEGF is also known for its suppressive role in inhibiting the differentiation and maturation of DCs [58]. Therefore, strategies aiming at combining anti-VEGF related signaling with ICI therapy could be the ideal regimen to further overcome the immunosuppressive nature of the TME by inducing tumor vascular normalization as well as enhancing DC maturation, optimizing the treatment efficacy of ICI therapy. Recently, two clinical trials that used anti-VEGF and anti-VEGFR2 antibodies, respectively, as add-on therapies to ICI monotherapy in HCC have shown encouraging results (Table 2). In IMbrave150, a global, randomized, open-label phase III clinical trial, 501 patients were randomized at a 2:1 ratio to receive a standard dose of atezolizumab (1200 mg) followed by a high dose (15 mg/kg) of bevacizumab (anti-VEGF antibody) every three weeks, or sorafenib. This trial did not include atezolizumab or bevacizumab monotherapy [51]. This study met its primary end points (both PFS and OS) at the first interim analysis after a median follow-up of 8.6 months, and therefore stopped. Survival in the patient arm receiving atezolizumab plus bevacizumab still had not been reached at the time the first interim ended; however, improved OS (HR 0.58; 95% CI 0.42–0.79; *p* = 0.00006) and improved PFS (HR 0.59; 95% CI 0.47–0.76; *p* < 0.00001) were observed. It was the first time that a phase III trial study had outperformed sorafenib in a first-line setting. Furthermore, updated OS data from IMbrave150 showed even more promising results. The median OS was 19.2 months in the combination therapy arm versus 13.4 months in the sorafenib arm. A survival benefit with combination therapy over sorafenib was observed; 52% of patients who received atezolizumab plus bevacizumab and 40% of patients who received sorafenib had survived after 18 months. The updated ORR in the combination therapy arm was 29.8% (95% CI 24.8–35.0) according to RECIST 1.1 criteria, which was superior to that in the sorafenib arm (11.3%; 95% CI 6.9–17.3), and 18 patients (5.5%, 18/326 patients) in the atezolizumab plus bevacizumab group had a complete response, as compared to no patients in the sorafenib group. Notably, patient health-related quality of life was also significantly improved. A longer median time to deterioration was observed in the combination treatment arm than in the sorafenib arm, 11.2 months versus 3.6 months (HR 0.63, 95% CI 0.46–0.85), respectively [59]. More recently, a similar result was also obtained from a phase II/III trial in China testing a combination of another ICI, sintilimab, and IBI305, a bevacizumab biosimilar, versus sorafenib as a front-line treatment for patients with unresectable HBV-associated HCC [48]. This study also proved that patients with combination treatment (median OS: not reached) also survived much longer than patients with sorafenib (median OS: 10.4 months). Due to the lack of a single-agent arm in the above studies, it is uncertain whether anti-PD1 or anti-VEGF therapy contributed more to the survival benefit of the combination treatment. Nevertheless, the previous phase 1b study showed that the response rates and PFS of the atezolizumab plus bevacizumab arm were significantly better than the bevacizumab treatment arm, suggesting that anti-PD1 may be an indispensable factor in designing future combined regimens for HCC [60]. 

### 3.4. Potential Novel Checkpoint Inhibitors

Aside from PD1/PDL1 and CTLA-4, here we discuss other important co-inhibitory molecules expressed by T cells which have been identified as potential immune checkpoints that modulate T cell activation. T-cell immunoglobulin and mucin-domain containing-3 (TIM3) is a membrane-bound protein that is originally expressed in CD8^+^ cytotoxic T cells and interferon-γ-producing CD4^+^ T helper 1 (Th1) cells [65]. Initially, the function of TIM3 was associated with autoimmune disease. Blocking of TIM3 with TIM3-specific antibody or administration of TIM-3 immunoglobulin (Ig) fusion protein resulted in Th1 cell and macrophage hyperactivation in an experimental autoimmune encephalomyelitis (EAE) mouse model [66]. Later, TIM3 expression was found in several immune cells, including T_reg_ cells [67], myeloid cells [68], natural killer (NK) cells [69] and dendritic cells (DCs) [65]. Notably, co-expression of TIM3 and PD1 has been observed in dysfunctional Th1 cells that express low levels of IL-2 and IFN-γ in the preclinical model [70]. A synergistic anti-tumor effect of combined anti-PD1 and anti-TIM3 immunotherapy has been observed in solid tumors in mouse models [70,71]. Therefore, combination treatment of TIM3 and PD1 or CTLA-4 inhibitor could serve as an attractive ICI regimen for HCC. Recently, a phase II trial assessing the efficacy and safety of anti-PD1 and anti-TIM3 combination therapy (NCT03680508) has been launched; the results are still being awaited. 

Another potential immune checkpoint is lymphocyte-activation gene 3 (LAG3). The structure of LAG3 is closely related to CD4, and they share the same ligand, MHC-II [72]. LAG3 expression is found on activated CD4^+^ T cells [73], CD8^+^ cells [74], T_reg_ cells [75] and plasmacytoid dendritic cells (pDCs) [76]. High levels of LAG3 have been found in tumor-infiltrating lymphocytes (TILs) [77]. Unlike PDL1, high expression of LAG3 has been found in several solid tumors, including HCC [78]. LAG3 also has been shown to have a negative impact on autoimmunity. Loss of LAG3 significantly accelerates disease progression of type 1 diabetes in Non-Obese-Diabetic (NOD) mouse models [79], and is associated with autoimmune disease. Importantly, interplay between LAG3 and other immune checkpoints has been noted. LAG3 could work cooperatively with PD1 or CTLA-4 to suppress MHC-II-mediated T cell receptor (TCR) signaling and subsequent T cell action, making it an ideal target for ICI therapy [78,80]. In an ovarian cancer model, a synergistic anti-tumor effect of combinatory treatment of LAG3 and PD1 mAb shed light on future potential combination therapies [80]. Several clinical trials are ongoing to evaluate the clinical benefits of LAG3 mAb and a soluble LAG3 Ig as well as combination therapy using anti-LAG3 and anti-PD1 mAb in an array of human malignancies [81,82]. Given that upregulation of LAG3 has been found in tumor-infiltrating CD8^+^ T cells and HCC cells [78,83], potential clinical trials for HCC can be expected in the near future. 

Another important aspect to examine is that compensatory upregulation of TIM3 and LAG3 may confer resistance to anti-PD1/PDL1 treatment. In a preclinical mouse model, Oweida et al. demonstrated that TIM3 upregulation was observed in tumor-infiltrating CD4 and CD8 T cells in murine head and neck squamous cell carcinoma (HNSCC) tumors treated with radiotherapy (RT) and anti-PDL1 therapy. Combined treatment with anti-TIM3, anti-PDL1 and RT led to the promotion of T cell cytotoxicity, decreased T_reg_ and significant tumor regression as compared with anti-PDL1 plus RT treatment [84]. In an effort to address the resistance mechanisms of PD1 blockade, Limagne et al. showed that accumulation of galetin-9-expressing monocytic MDSCs (mMDSCs) and TIM3-expressing CD8 T cells was found in lung cancer patients with resistance to anti-PD1 therapy, and may play a crucial role in resistance to PD1 blockade. They demonstrated that anti-TIM3 antibody in vitro could reverse resistance to anti-PD1 in peripheral blood mononuclear cells (PBMCs) isolated from lung cancer patients. Moreover, galetin-9-expressing mMDSCs could impede TIM3+ CD8 T cell activity to reduce anti-PD1 treatment efficacy [85]. In addition, Jikova et al. analyzed fresh HCC biopsies and peripheral blood samples from 21 HCC patients treated with sorafenib or PD1/PDL1 blockade therapy to show that non-responders tended to have TIM3 and LAG3 upregulation on circulating T cells compared with responders [86]. 

Taken together, blocking the compensatory upregulation of other checkpoint inhibitors such as TIM3 and LAG3 after anti-PD1/PDL1 treatment could be an important pharmaceutical strategy to overcome primary and secondary resistance to PD1/PDL1 blockade in patients with advanced HCC in the future. 

## 4. Challenges and Opportunities in ICI Therapy for HCC

### 4.1. Management of ICI-Related Immune-Mediated Adverse Effects (IMAEs)

Inhibitory checkpoint molecules expressed by immune cells are to intended to serve as a brake for immune cell overactivation. Therefore, blocking the inhibitory functions of checkpoint molecules will inevitably lead to developing a range of immune-mediated adverse events (IMAEs) including hepatitis, colitis pneumonitis, etc. [87]. The detailed management of autoimmune-specific IMAEs in patients with ICIs therapy has been previously reviewed by Brahmer, et al. [88]. Although HCC usually develops from a background of chronic inflammation induced by HBV/HCV or alcohol, the overall incidence of IMAE in HCC patients receiving ICI therapy is similar to those of patients with other cancer types [89]. Currently, the majority of IMAEs caused by ICI therapy can be relieved by administration of corticosteroids in combination with mycophenolate mofetil or azathioprine [89]. Here, we summarize latest safety profiles of ICIs for HCC (Table 2). As compared to sorafenib treatment, HCC patients receiving nivolumab treatment not only had more durable disease control (median 7.5 months vs. 5.7 months) but also had a better safety profile, with fewer treatment-related adverse effects (TRAES) over grade 3 (22% vs. 49%). However, the use of anti-CTLA-4 agents has often seen more serious IMAEs than that of anti-PD1/PL1 agents. Dose-related adverse effects were frequently observed in HCC patients, causing treatment discontinuation [45]. In the CheckMate040 phase II trial, 50% of patients were treated with corticosteroids in the anti-CTLA-4 high dose group while only 24% in low dose group received corticosteroids [42]. This phenomenon was also observed to a lesser extent in the treatment of patients with other cancer types, such as melanoma and lung cancer [90]. On the other hand, safety profiles between anti-PD1 and anti-PDL1 therapy were comparable [91]. Dual blockade of PD1 and CTLA-4 displayed superior treatment efficacy to each monotherapy alone; however, it came at the cost of an increased rate of hepatic TRAE in the early phase of treatment. Fortunately, the conditions were improved in most of patients after six weeks, suggesting no signs of synergistic toxicities [45]. Overall, PD1/PDL1 inhibitors result in less serious TRAE than CTLA-4 inhibitors. Therefore, dosing and timing for anti-CTLA-4 therapy seems critical in treating patients with HCC.

### 4.2. Possible Predictive Factors for Response to Anti-PD1/PDL1 Therapy

Given that the action mechanism of PD1/PDL1 inhibitors is to block PD1/PDL interaction between tumor cells and immune cells (Figure 1), PDL1 expression status has served as a biomarker for predicting the treatment outcomes of patients receiving anti-PD1/PDL1 therapy. Recent meta-analysis indicates that a survival benefit from ICI therapy was observed in cancer patients with the up-regulation of PDL1 expression (PDL1 > 1% score index), but not in those with <1% PDL1 [92]. Survival benefit between patients with >1% and ≥50% PDL1 was slightly improved, but not statistically significant. Therefore, PDL1 expression status must be taken into account in the context of ICI therapy. 

Interestingly, this seems to not be the case in the recent ICI clinical trials for advanced HCC. As indicated in KEYNOTE-240 phase III trial, HCC patients with PDL1 expression in either tumor cells or stroma cells had a similar response to pembrolizumab treatment as compared to those with no PDL1 expression (median survival 16.1 months versus 16.7 months) [44]. In the CheckMate 040 trial, tumor responses were also independent of PDL1 status [93]. Other cellular factors and immune signatures that could serve as biomarkers to predict ICI immunotherapy outcomes were summarized previously [94].

## 5. Conclusions

At one time, the introduction of ICIs in advanced HCC was far behind the other human solid tumors. However, with the substantial exciting data produced from recent checkpoint blockade clinical trials in advanced HCC, atezolizumab plus bevacizumab combinatorial treatment is now approved by FDA as the frontline standard of care in advanced HCC [95,96], and the availability of new ICI combination regimens can be expected to significantly increase the treatment efficacy of regional therapy and neoadjuvant therapy in patients with unresectable disease. Although advanced HCC is still among the most difficult-to-treat human cancers, the emergence of immune checkpoint blockade therapies has drastically changed the landscape of clinical treatment of advanced HCC. The potential novel immune checkpoint inhibitors under development will further add to the repertoire of ICI therapy. The challenge of treating patients with cirrhosis remains a difficult task. Nevertheless, treatment of this seemingly incurable disease is now within reach with ICIs, which offer the best hope of reducing the mortality rates of HCC. The quest to eradicate this deadly disease is only beginning.

## Figures and Tables

**Figure 1 cancers-13-05295-f001:**
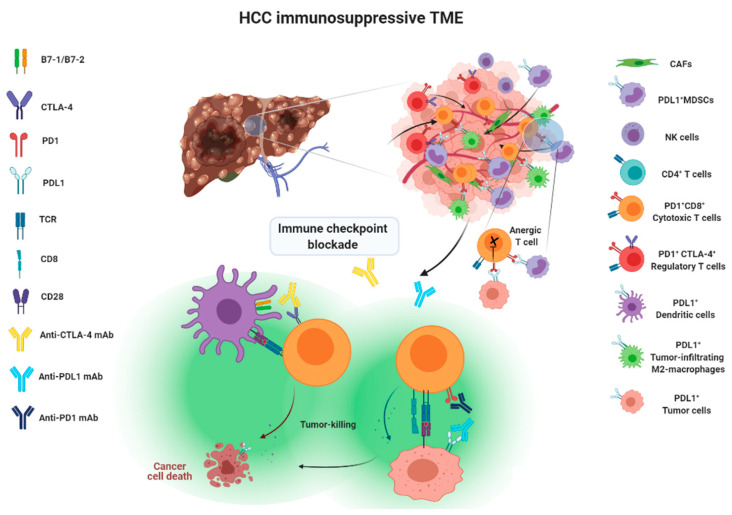
Action mechanisms of immune checkpoint blockade therapy.

**Figure 2 cancers-13-05295-f002:**
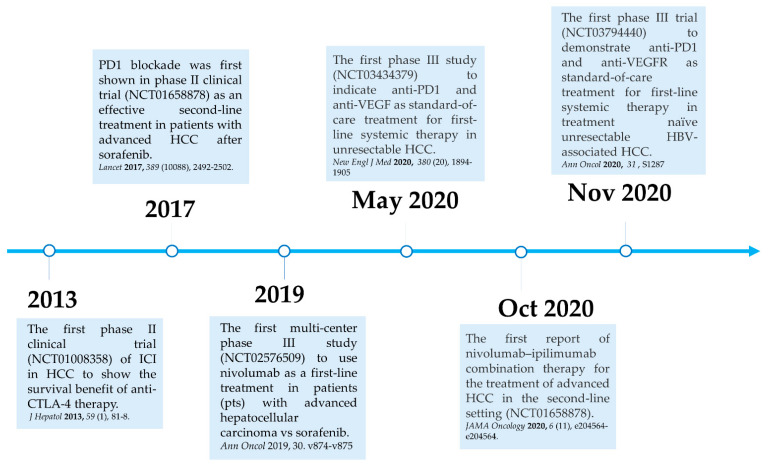
Milestones for immune checkpoint therapy for patients with advanced HCC.

**Table 1 cancers-13-05295-t001:** Current phase III trials involving ICI monotherapy and combinations of ICIs and TKI or VEGF inhibitors.

Trial Identifier	ICI/Isotype	Drug	Treatment Arm	Eligible Patient/Setting	Endpoint	Ref.
Monotherapies						
NCT02702401	PD1/IgG4	Pembrolizumab	1. Placebo 2. Pembrolizumab	Advanced HCC/2L	Dec. 2022	[44]
NCT02576509	PD1/IgG4	Nivolumab Sorafenib	1. Sorafenib 2. Nivolumab	Advanced HCC/1L	June 2021	[45]
NCT03755739	PD1/IgG4	Pembrolizumab	1. Peripheral in fusion	Advanced HCC/1L	Nov. 2021	Ongoing
			2. Artery infusion			
			3. Intra-tumor infusion			
NCT03062358	PD1/IgG4	Pembrolizumab	1. Placebo 2. Pembrolizumab	Advanced HCC/2L	Jan. 2022	Ongoing
NCT03412773	PD1/IgG4	Tislelizumab	1. Placebo 2. Tislelizumab	Unresectable HCC/1L	May 2022	[46]
Combination therapies of ICI and TKI or anti-angiogenic agents						
NCT03298451	PD1 CTLA-4 (IgG4)	Durvalumab Tremelimumab	1. Sorafenib 2. Tremelimumab 3. Tremelimumab plus durvalumab	HCC BCLC stage B not eligible for locoregional therapy/1L	June 2021	[47]
NCT03794440	PD1/IgG4 VEGF/IgG1	Sintilimab Bevacizumab biosimilar	1. Sorafenib 2. Sintilimab plus Bevacizumab biosimilar	Advanced HCC/1L	Dec. 2022	[48]
NCT03847428	PDL1/IgG1 VEGF/IgG1	Durvalumab Bevacizumab	1. Combination with resection/MWA 2. Resection/MWA alone	HCC eligible for curative resection/MWA/2L	June 2023	[49]
NCT03713593	PD1/IgG4 VEGFR	Pembrolizumab Lenvatinib	1. Lenvatinib 2. Pembrolizumab	Advanced HCC/1L	July 2022	[50]
NCT03764293	PD1/IgG4 TKI	Camrelizumab Apatinib	1. Apatinib	Advanced HCC/1L	Jan. 2022	Ongoing
NCT03434379	PDL1/IgG1 VEGF/IgG1	Atezolizumab Bevacizumab	1. Sorafenib 2. Atezolizumab plus Bevacizumab	Advanced HCC/1L	June 2022	[51]

HCC: Hepatocellular carcinoma; BCLC: Barcelona Clinic Liver Cancer; 2L: second-line therapy; 1L: first-line therapy; CTLA-4: Cytotoxic T-lymphocyte-associated antigen 4; MWA: Microwave ablation; PD1: Programmed cell death protein 1; VEGF: Vascular endothelial growth factor; VEGFR: vascular endothelial growth factor receptor.

**Table 2 cancers-13-05295-t002:** Safety profiles of immune checkpoint inhibitors for HCC.

PD1/PDL1	Combination Agent	TRAE (%)				
		Total	Grade ≥ 3	Treatment Discontinuation	Severe	Ref.
Monotherapies						
Nivolumab		83	25	6	6	[42]
Pembrolizumab (200 mg every 3 weeks)		73	26	17	15	[43]
Camrelizumab (3 mg/kg every 2 or 3 weeks)		NR	22	4	11	[61]
Durvalumab (1500 mg every 4 weeks)		60	20	8	11	[62]
Tremelimumab (750 mg every 4 week)		84	43	13	25	[62]
Atezolizumab (1200 mg every 3 weeks)		41	5	2	3	[60]
Combination therapies of two immune checkpoint inhibitors						
Durvalumab (1500 mg every 4 weeks)	Tremelimumab (300 mg single dose)	82	35	11	16	[62]
Durvalumab (1500 mg every 4 weeks)	Tremelimumab (75 mg every 4 weeks ×4)	69	24	6	14	[62]
Nivolumab (3 mg/kg every 3 weeks)	Ipilimumab (1 mg/kg every 3 weeks)	71	29	6	18	[45]
Nivolumab (1 mg/kg every 3 weeks)	Ipilimumab (3 mg/kg every 3 week)	94	53	22	22	[45]
Combination therapies of an immune checkpoint inhibitor and a TKI						
Pembrolizumab (200 mg every 3 weeks)	Lenvatinib (8 or 12 mg/day)	94	80	10	59	[63]
Nivolumab (240 mg every 2 weeks)	Cabozantinib (40 mg/day)	89	47	NR	NR	[64]
Nivolumab (240 mg every 2 weeks)	Ipilimumab (1 mg/kg every 6 weeks plus cabozantinib 40 mg/day)	94	71	15.5	NA	[64]
Combination therapies of an immune checkpoint inhibitor and anti-VEGF agent						
Atezolizumab (1200 mg every 3 weeks)	Bevacizumab (15 mg/kg every 3 weeks)	88	39	NR	24	[60]
Atezolizumab (1200 mg every 3 weeks)	Bevacizumab (15 mg/kg every 3 weeks)	84	98	15	17	[51]

TRAE: Treatment-related adverse effect; HCC: Hepatocellular carcinoma; VEGF: vascular endothelia growth factor; TKI: Tyrosine kinase inhibitor; NR: Not reported.

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
