# Peer review of "Cure the Incurable? Recent Breakthroughs in Immune Checkpoint Blockade for Hepatocellular Carcinoma"

_cancers, 2021, doi:10.3390/cancers13215295_

Round 1

Reviewer 1 Report

This is well put together .A few comments

  1. Table 1  does not specify whether  first or second line treatments 
  2. Bevacizumab  may also work in synergy with ICIs via a vasculature normalisation  effect.This should be described and referenced 
  3. The authors shown tone down comments re the Checkmate and Keynote studies as they did not show improved overall survival or primary endpoints and hence unlikely to become standard regimes 
  4. Actual autoimmune  specific  AEs should be referenced with management links in an additional   summary table

Author Response

Dear reviewer 1,

Please see the attachment for a point-by-point response and the revised manuscript.

A point-by-point response is attached at the end of the manuscript.

Best regards,

Shih-Hsuan Chan

Reviewer 2 Report

Title: Cure the incurable? Recent breakthroughs in immune checkpoint blockade for Hepatocellular Carcinoma

In this manuscript, the authors provide extensive information about the history of ICIs therapy in the HCC field, up-to-date information for the latest ICIs clinical trials, and discuss the recent development of novel therapeutical strategies that would potentially lead to a new immune checkpoint blockade in the field of advanced HCC. 

It is a very interesting and attractive review for both clinicians and basic researchers. The manuscript is very well written and only minor modifications are requested. 

  • Part „Simple Summary“ can be improved. Mainly the sentence „Due to patients are usually diagnosed at an advanced stage, and therefore there are very few treatment options left plus lack of effective therapies, the clinical outcome of patients with advanced HCC are very poor“ should be revisited.
  • I would propose to slightly modify the sentence „As mentioned before, VEGF secreted by cancer cells is important to foster immuno suppressive TME not because for its role in recruiting endothelial cells to promote tumor angiogenesis but for its suppressive role in inhibiting differentiation and maturation of DCs [57]“ as following „VEGF secreted by cancer cells is important to foster immunosuppressive TME not because for its role in recruiting endothelial cells to promote tumor angiogenesis but mainly for its suppressive role in inhibiting differentiation and maturation of DCs [57]“
  1. The information about complete responses can be added in „3.2. Combination therapy and 3.3. Combination therapy of ICIs and VEGF inhibitor“ part.
  2. Compensatory upregulation of other inhibitory checkpoint molecules after anti-PD-1/PD-L1 treatment should be briefly discussed in “3.4. Potential novel Checkpoint inhibitors“ to explain clearly why targeting other immune checkpoints is a very important pharmaceutical strategy. For instance, in both preclinical and clinical studies, the upregulation of TIM3 was detected on TILs after treatment, leading to acquired resistance in PD1/PDL1 blockade (Oweida et al., 2018; Limagne et al., 2019). In HCC patients that were non-responders to anti-PD-1/PDL1 therapy, the compensatory upregulation of TIM3 and LAG3 was observed https://doi.org/10.14309/ctg.0000000000000058
  3. I would recommend renaming chapter „4.2. Patients with advanced HCC receiving anti-PD1/PDL1 therapy show survival benefit regardless of PDL1 expression status“ to „4.2. Possible predictive factors for response to anti-PD1/PDL1 therapy“ It should be also mentioned that other possible factors were summarized previously here: doi:10.3390/cancers11101554 ; https://doi.org/10.2147/JHC.S322289
  4. The paragraph 343-348 - „For once, the introduction of ICIs in advanced HCC was far behind the other human solid tumors. But with the substantial exciting data produced from recent checkpoint blockade clinical trials in advanced HCC, atezolizumab plus bevacizumab combinatorial treatment is now approved by FDA as the frontline standard-of-care in advanced HCC [90, 91] and the availability of new ICI combination regimens would expect to significantly increase the treatment efficacy of regional therapy as well as neoadjuvant therapy in patients with unresectable disease.“ - should be moved to "Conclusions" as there is no link with a previous chapter focused on PDL1 status.

Author Response

Dear reviewer 2,

Please see the attachment for a point-by-point response and the revised manuscript.

A point-by-point response is attached at the end of the manuscript.

Best regards,

Shih-Hsuan Chan
